# Growth and Physiological Response of *Viola tricolor* L. to NaCl and NaHCO_3_ Stress

**DOI:** 10.3390/plants12010178

**Published:** 2023-01-01

**Authors:** Xiaoe Liu, Shiping Su

**Affiliations:** College of Forestry, Gansu Agricultural University, Lanzhou 730070, China

**Keywords:** *Viola tricolor* L., salt stress, growth, physiological response, tolerance limit

## Abstract

Soil salinization is an important environmental problem worldwide and has a significant impact on the growth of plants. In recent years, the mechanisms of plant salt tolerance have received extensive attention from researchers. In this paper, an experiment was implemented to assess the potential effect of different NaCl and NaHCO_3_ (sodium bicarbonate—an alkaline salt) concentrations (25 mmol·L^−1^, 50 mmol·L^−1^, 100 mmol·L^−1^, 150 mmol·L^−1^ and 200 mmol·L^−1^) on the growth, antioxidant enzymes, osmoprotectants, photosynthetic pigments and MDA of *Viola tricolor* L. to reveal the physiological response and explore the maximum concentrations of NaCl and NaHCO_3_ stress that *V. tricolor* can tolerate. The results showed that NaCl and NaHCO_3_ treatments had significant effects on osmoprotectants, antioxidant enzymes, photosynthetic pigments, MDA content and the plant height growth of *V. tricolor*. On day 14 of the NaCl and NaHCO_3_ stress, the height growth of *V. tricolor* was significantly greater than CK when the concentration of NaCl and NaHCO_3_ was less than 100 mmol·L^−1^. Soluble protein (SP) was significantly greater than CK when the NaCl concentration was less than 150 mmol·L^−1^ and the NaHCO_3_ concentration was less than 200 mmol·L^−1^; soluble sugar (SS) was significantly greater than CK under all NaCl and NaHCO_3_ treatments; proline (Pro) was significantly greater than CK when the NaCl concentration was 150 mmol·L^−1^ and the NaHCO_3_ concentration were 150 and 200 mmol·L^−1^, respectively. Peroxidase (POD) was significantly greater than CK when the NaCl concentration was less than 200 mmol·L^−1^ and the NaHCO_3_ concentration was less than 150 mmol·L^−1^; superoxide dismutase (SOD) was significantly greater than CK when the NaCl concentration was 50 mmol·L^−1^ and the NaHCO_3_ concentrations were 50, 100 and 150 mmol·L^−1^, respectively; catalase (CAT) was significantly greater than CK when the NaCl and NaHCO_3_ concentrations were 25, 50 and 100 mmol·L^−1^, respectively. Chlorophyll (Chl) was significantly lower than CK when the NaCl and NaHCO_3_ concentrations were greater than 100 mmol·L^−1^. Malondialdehyde (MDA) gradually increased with the increase in the NaCl and NaHCO_3_ concentrations. Membership function analysis showed that the concentrations of NaCl and NaHCO_3_ that *V. tricolor* was able to tolerate were 150 mmol·L^−1^ and 200 mmol·L^−1^, respectively. Beyond these thresholds, osmoprotectants and antioxidant enzymes were seriously affected, Chl degradation intensified, the photosynthetic system was seriously damaged, and the growth of *V. tricolor* was severely affected. According to a comprehensive ranking of results, the degree of NaCl stress on *V. tricolor* was lower than that from NaHCO_3_ when the treatment concentration was lower than 50 mmol·L^−1^, but higher than that from NaHCO_3_ when it exceeded 50 mmol·L^−1^.

## 1. Introduction

Land salinization is a significant environmental problem. There are approximately 1 billion hm^2^ of saline-alkali land worldwide, and it is growing at a rate of 1–1.5 million hm^2^ per year [1,2]. In China, there are approximately 36 million hm^2^ of saline-alkali land, accounting for 4.88% of the available land area [2], and it is estimated that about 50% of the available cultivated land will be threatened by salinization by 2050 [3]. Several lines of research have demonstrated that physiological and growth indices of plants are affected by salt stress. Osmoprotectants, such as proline (Pro), soluble sugar (SS) and soluble protein (SP), increase. Malondialdehyde (MDA) also showed an increasing trend [4,5,6,7], while the activities of antioxidant enzymes, such as superoxide dismutase (SOD), peroxidase (POD) and catalase (CAT) increase under low NaCl concentrations and gradually decrease with an increase in NaCl concentration [8,9]. The content of the photosynthetic pigment increased at first and then decreased with an increase in the NaCl concentration [10], thus affecting various aspects of plant growth, such as a reduction in plant height [11], restriction of root growth [12], reduction in biological leaves area and yield, and the occurrence of necrosis spots [13,14].

The saline-alkali land is mainly dominated by carbonate and chloride salts, such as NaCl and NaHCO_3_ in northwest China, which seriously affect the normal growth of plants. However, different kinds of salt stress on plants have different mechanisms, and the tolerance of plants to salt stress varies with the salt type and concentration [15]. When plants are subjected to NaCl stress, the structure and function of the cell membrane are altered owing to the dual stress imposed by Na^+^ and Cl^−^. The original Ca^2+^ on the cell membrane is replaced by Na^+^, resulting in tiny holes in the cell membrane and ion leakage, which leads to changes in the type and concentration of ions in the cell; a large amount of salt accumulation in the cell leads to solidification of the protoplasm and chlorophyll (Chl) destruction. Hence, the rate of photosynthesis decreases sharply, which seriously affects the growth and development of plants [10]. Meanwhile, the stress imposed by alkaline salts such as sodium bicarbonate (NaHCO_3_) manifests itself not only as ion toxicity and osmotic stress, but also as the stress related to high pH. The increase in pH leads to a decrease in the availability of metal ions such as Fe^2+^, Mn^2+^, Mg^2+^ and Ca^2+^, which in turn leads to plant physiological and metabolic disorders and the inhibition of growth [16]. Research has shown that under NaCl and NaHCO_3_ stress with equal concentrations of Na^+^, the SP, free proline and Pro under NaHCO_3_ stress were significantly higher than those under NaCl stress, because the higher pH under the NaHCO_3_ stress was more likely to cause degradation of photosynthetic pigments and damage to photosynthetic apparatus, resulting in reduced photosynthetic function [17].

*Viola tricolor* L. is an annual or subshrub of the Violaceae family. It prefers to grow in weakly acidic, neutral or clay loam soil with a pH of 5.4–7.4 which is well drained and rich in organic matter. It is often used for flowerbeds, flower pools, flower mirrors and patterned flowerbeds in landscaping. The whole of the plant can be used in medicine, with various functions such as lowering the temperature, detoxification, dispersing blood stasis and relieving coughs [18]. Studies have also shown that *V. tricolor* flowers have high nutritional value because they are rich in anthocyanins, flavonoids and carotenoids [19]. Therefore, research on the cultivation and stress resistance of *V. tricolor* has been receiving increasing attention. At present, *V. tricolor* has been cultivated in a large area in China. However, as one of the countries with the largest populations in the world, China needs a large area of cultivated land to grow crops in order to ensure national food security; therefore, there is not enough cultivated land to grow other plants such as *V. tricolor*. However, there are large areas of salinized land in China, so planting *V. tricolor* on salinized land will be the development trend, which can not only improve the sustainable utilization rate of salinized land, but also realize the medicinal, edible and ornamental value of *V. tricolor*.

To date, research on *V. tricolor* has been carried out in areas of drought stress [20], salt stress [21,22], temperature stress [23,24,25] and heavy metal stress [26]. You Yang et al. and Liu Huichao et al. found that under the stress of 0–250 mmol·L^−1^ NaCl, SOD, POD, CAT and chlorophyll of *V. tricolor* showed a trend of first increasing and then decreasing [21,22]. But there is no research on osmoregulation substances, malondialdehyde, growth and physiological changes of *V. tricolor* under neutral salt and alkaline salt stress. Meanwhile, the tolerance limit of plants to neutral salt and alkaline salt stress is the result of the comprehensive action of various physiological and biochemical factors, and all relevant factors need to be comprehensively evaluated. Therefore, we hypothesize that the response trend of *V. tricolor* to NaCl and NaHCO_3_ stress in growth, physiological and biochemical characteristics is consistent, but the stress degree caused by the same concentration of NaCl and NaHCO_3_ stress to *V. tricolor* is different, while the highest tolerance concentration of *V. tricolor* to NaCl and NaHCO_3_ stress is different. Therefore, the main objective of the current study was to investigate the response mechanisms of osmoprotectants, antioxidant enzymes, photosynthetic pigments and malondialdehyde (MDA) in *V. tricolor* leaves, as well as their growth, under NaCl and NaHCO_3_ stress, as well as to determine the maximum soil NaCl and NaHCO_3_ contents that *V. tricolor* can tolerate. This will provide a theoretical basis for the cultivation of *V. tricolor* in saline-alkali soil.

## 2. Results

### 2.1. Effects of NaCl and NaHCO_3_ on Osmoprotectants in V. tricolor Leaves

With the increase in NaCl and NaHCO_3_ concentration, soluble protein (SP) increased first and then decreased on days 7 and 14 of the treatments (Figure 1). On day 14, when the concentration of NaCl and NaHCO_3_ was 50 mmol·L^−1^, the SP was the highest, having increased by 96.79% and 140.86% compared with CK, respectively, and there were significant differences with CK (*p* < 0.05). When the concentrations of NaCl and NaHCO_3_ were 200 mmol·L^−1^, the SP increased by −14.65% and 58.15% compared with CK, respectively. The SP under NaHCO_3_ treatments was higher than that under NaCl treatments (Figure 1). Under NaCl treatments, the SP on day 14 was lower than that on day 7; while under NaHCO_3_ treatments, the SP on day 14 was higher than that on day 7 at a low concentration (lower than 50 mmol·L^−1^) and lower than that on day 7 at a high concentration (higher than 50 mmol·L^−1^). Except for the 200 mmol·L^−1^ NaCl treatment, the SP in other treatments was higher than that in CK, and its range of increase compared with CK showed a decreasing trend with the extension of the treatment time (Figure 1).

With the increase in NaCl and NaHCO_3_ concentration, soluble sugar (SS) and proline (Pro) increased continuously on day 7 of the treatments, and increased first and then decreased on day 14 of the treatments (Figure 1). On day 14, when the concentrations of NaCl and NaHCO_3_ were at 50 mmol·L^−1^, the SS was the highest, increasing by 427.65% and 450.39% compared with CK, respectively, and there were significant differences with CK (*p* < 0.05). When the concentrations of NaCl and NaHCO_3_ were at 200 mmol·L^−1^, the SS increased by 228.38% and 338.91% compared with CK, respectively, and there were significant differences with CK (*p* < 0.05) (Figure 1). On day 14, the SS under NaHCO_3_ treatments was higher than that under NaCl treatments. Except for the 200 mmol·L^−1^ NaCl and NaHCO_3_ treatments, the SS on day 14 was higher than that on day 7, while the SS under the other treatments was significantly higher than that of CK (*p* < 0.05), and its range of increase compared with CK showed a decreasing trend with the increase in treatment concentration (Figure 1).

When NaCl and NaHCO_3_ concentrations were at 150 mmol·L^−1^, Pro was the highest, increasing by 121.08% and 112.12% compared with CK, respectively, and there were significant differences with CK (*p* < 0.05). When the concentrations of NaCl and NaHCO_3_ were 200 mmol·L^−1^, Pro increased by 28.46% and 89.26% compared with CK, respectively, and there were significant differences with CK (*p* < 0.05). Except for the 150 mmol·L^−1^ treatment, Pro under NaHCO_3_ treatments was higher than that under NaCl treatments; and except for 25 mmol·L^−1^ NaCl treatment, Pro on day 14 was lower than that on day 7 with the extension of the treatment time (Figure 1).

### 2.2. Effects of NaCl and NaHCO_3_ on Antioxidant Enzyme Activity in V. tricolor Leaves

With the increase in NaCl and NaHCO_3_ concentrations, peroxidase (POD) increased first and then decreased on days 7 and 14 of the treatments (Figure 2). On day 14, when the NaCl and NaHCO_3_ concentrations were at 50 mmol·L^−1^, POD was the highest, increasing by 89.60% and 108.07% compared with CK, respectively, and there were significant differences with CK (*p* < 0.05). When the concentrations of NaCl and NaHCO_3_ were at 200 mmol·L^−1^, POD decreased by 8.33% and 3.31% compared with CK, respectively, but there were no significant differences with CK (*p* > 0.05) (Figure 2). When the NaCl and NaHCO_3_ concentrations were lower than 50 mmol·L^−1^, the POD under NaHCO_3_ treatments was higher than that under NaCl treatments. POD on day 14 was higher than that on day 7 (Figure 2).

With the increase in NaCl and NaHCO_3_ concentration, superoxide dismutase (SOD) increased first and then decreased on day 7 and day 14 of the treatments (Figure 2). On day 14, when the NaCl concentration was 50 mmol·L^−1^ and the NaHCO_3_ concentration was 100 mmol·L^−1^, SOD was the highest, increasing by 52.41% and 65.18% compared with CK, respectively (*p* < 0.05). When the concentrations of NaCl and NaHCO_3_ were 200 mmol·L^−1^, SOD decreased by 11.91% and 1.75% compared with CK, respectively (Figure 2). When NaCl and NaHCO_3_ concentrations were higher than 100 mmol·L^−1^, SOD under NaHCO_3_ treatments was higher than that under NaCl treatments. Except for the 100 mmol·L^−1^ NaHCO_3_ treatment, the SOD on day 14 was lower than that on day 7 (Figure 2).

With the increase in NaCl or NaHCO_3_ concentration, catalase (CAT) increased first and then decreased on day 7 and 14 of the treatments (Figure 2). On day 14, when the NaCl concentration was at 100 mmol·L^−1^ and the NaHCO_3_ concentration was at 50 mmol·L^−1^, CAT was the highest, increasing by 319.39% and 156.96% compared with CK, respectively, and there were significant differences with CK (*p* < 0.05). When the concentrations of NaCl and NaHCO_3_ were at 200 mmol·L^−1^, CAT increased by −17.97% and 74.84% compared with CK, respectively (Figure 2). When the NaHCO_3_ and NaCl concentrations were lower than 100 mmol·L^−1^, the CAT on day 14 was higher than that on day 7 (Figure 2).

### 2.3. Effects of NaCl and NaHCO_3_ on Photosynthetic Pigment in V. tricolor Leaves

With the increase in NaCl and NaHCO_3_ concentration, the photosynthetic pigment increased first and then decreased on day 7 and day 14 (Figure 3). On day 14, when the NaCl concentration was at 50 mmol·L^−1^ and the NaHCO_3_ concentration was at 25 mmol·L^−1^, the photosynthetic pigment was the highest, increasing by 6.50% and 7.92% compared with CK, respectively (*p* < 0.05). When the concentrations of NaCl and NaHCO_3_ were 200 mmol·L^−1^, the photosynthetic pigment decreased by 30.05% and 23.26% compared with CK, respectively, and there were significant differences with CK (*p* < 0.05) (Figure 3). The photosynthetic pigment increased significantly under 50 mmol·L^−1^ and 25 mmol·L^−1^ NaCl treatments as well as the 25 mmol·L^−1^ NaHCO_3_ treatment compared with CK. When the concentrations of NaHCO_3_ and NaCl were higher than 50 mmol·L^−1^, photosynthetic pigment on day 14 was lower than that on day 7, but the differences in photosynthetic pigment between day 7 and day 14 under the NaCl treatments were greater than those under NaHCO_3_ treatments (Figure 3).

### 2.4. Effects of NaCl and NaHCO_3_ on MDA in V. tricolor Leaves

Malondialdehyde (MDA) increased on days 7 and 14 with the increase in NaCl and NaHCO_3_ concentration (Figure 4). On day 14, when the NaCl and NaHCO_3_ concentrations were 200 mmol·L^−1^, MDA was the highest, increasing by 101.97% and 81.90% compared with CK, respectively (*p* < 0.05) (Figure 4). When the concentrations were lower than 50 mmol·L^−1^, the MDA under NaHCO_3_ treatments was higher than that under NaCl treatments; additionally, when concentrations were higher than 50 mmol·L^−1^, the MDA under NaHCO_3_ treatments was lower than that under NaCl treatments (Figure 4).

### 2.5. Effects of NaCl and NaHCO_3_ on the Height Growth of V. tricolor

When NaCl and NaHCO_3_ concentrations were lower than 50 mmol·L^−1^, the height growth of *V. tricolor* was significantly promoted, and the promoted effect was higher under NaCl treatments than under NaHCO_3_ treatments (Figure 5). On day 14, the net growth of the plant height under 25 mmol·L^−1^ treatments was greatest, being 1.06 cm and 0.99 cm, respectively, and were significantly different from those under CK (*p* < 0.05). Under 100 mmol·L^−1^ treatments, the net growth of the plant height was the same as that under CK, at 0.66 cm and 0.68 cm, respectively. When the concentration was higher than 100 mmol·L^−1^, the growth in the plant height was inhibited, and this effect increased with the increase in NaCl and NaHCO_3_ concentrations, with the level of inhibition with NaCl being higher than with NaHCO_3_. Under the 200 mmol·L^−1^ treatments, the net growth in the plant height decreased by 45.43% and 28.48% compared with CK, respectively (*p* < 0.05) (Figure 5).

### 2.6. Comprehensive Evaluation of the Effects of NaCl and NaHCO_3_ on Stress in V. tricolor

The NaCl tolerance of plants is not the result of a single factor, rather; it is a comprehensive action of many physiological and biochemical factors. Therefore, in order to accurately evaluate NaCl and NaHCO_3_ tolerance, as well as maximum NaCl and NaHCO_3_ tolerance of *V. tricolor*, the measured parameters were carried out by a membership function analysis (Table 1). The results showed that, with the increase in NaCl and NaHCO_3_ concentration, the average membership function score increased first and then decreased, and that NaCl and NaHCO_3_ stress gradually intensified. When the NaCl concentration was 150 mmol·L^−1^ and the NaHCO_3_ concentration was 200 mmol·L^−1^, the membership function scores were 0.344 and 0.338, respectively, which were higher than or equal to those under CK (0.338). However, when NaCl concentration was higher than 150 mmol·L^−1^ and the NaHCO_3_ concentration was higher than 200 mmol·L^−1^, the membership function score was lower than that under CK treatment (Table 1). This showed that, with the increase in NaCl and NaHCO_3_ concentration, the level of stress increased, the accumulation of metabolic regulators and the activity of antioxidant enzymes were badly affected, the degradation of Chl was intensified, the photosynthetic system was seriously damaged, and the growth of *V. tricolor* was seriously inhibited, which indicated that *V. tricolor* could no longer alleviate the effects of NaCl and NaHCO_3_ stress by regulating its physiological metabolism. Therefore, the concentration limits of *V. tricolor* under NaCl and NaHCO_3_ stress were 150 mmol·L^−1^ and 200 mmol·L^−1^, respectively. According to the comprehensive ranking of the results, when the NaCl and NaHCO_3_ concentration was less than 50 mmol·L^−1^, the degree of stress for *V. tricolor* under NaCl treatments was lower than that under NaHCO_3_ treatments; and when the concentration was higher than 50 mmol·L^−1^, the degree of stress under NaCl treatments was higher than that under NaHCO_3_ treatments (Table 1).

## 3. Materials and Methods

### 3.1. Experimental Materials

The *V. tricolor* seedlings were grown from seeds obtained from Lanxiang Gardening, Lanzhou, China.

The substrate (produced in Denmark) was sterilized with 50% (g·V^−1^) wettable carbendazim powder, sealed for five days, and put into a nutrition bowl (diameter × height = 10 cm × 15 cm) for later use. During sowing, the substrate in the nutrient bowl was watered thoroughly with distilled water, and two *V. tricolor* seeds were sown in each nutrient bowl. After the seeds germinated and grew two true leaves, one seedling was kept in each nutrient bowl, and NaCl and NaHCO_3_ stress was applied when the seedlings were five or six true leaves (seedling age of about 45 days).

### 3.2. Experimental Design

NaCl and NaHCO_3_ were applied at six concentrations of 0 mmol·L^−1^ (CK, distilled water), 25 mmol·L^−1^, 50 mmol·L^−1^, 100 mmol·L^−1^, 150 mmol·L^−1^ and 200 mmol·L^−1^. There were three replicates per treatment and 30 plants per replicate. The pH values of the NaCl and NaHCO_3_ treatments were 7.01 and 8.30, respectively. In order to avoid any immediate shock effect after applying NaCl and NaHCO_3_, *V. tricolor* seedlings needed to be pretreated; the treatment solution with a concentration higher than 25 mmol·L^−1^ was irrigated every two days until the set concentration was reached (Table 2). The day when all treatments reached their target concentration was set as the first day of the NaCl and NaHCO_3_ stress treatment [27]. The irrigation amount was twice the water holding capacity of the matrix to ensure that two-thirds of the treatment solution flowed out, so as to wash away the accumulated NaCl and NaHCO_3_ in the previous stage.

### 3.3. Samples and Determination

On day 7 and day 14 of the NaCl and NaHCO_3_ treatments, 10 plants were randomly selected for the determination of growth and physiological indices, as follows:

Physiological indices: 10 plants of *V. tricolor* were selected for each repetition, and three leaves of the same size were selected for each plant, totaling 30 leaves, and placed in liquid nitrogen for preservation and determination of physiological indicators. When each index was measured, samples were taken from each leaf with a 0.5 cm diameter hole punch; the mixed leaves were then taken as samples.

The methods employed, as described by Li [28] and Gao [29], were as follows: SS was determined by anthronecolorimetry; Pro was determined by sulfosalicylic acid extraction; SP was determined by Coomassie brilliant blue staining; SOD was determined by nitrobluetetrazole photochemical reduction; POD was determined by the guaiacol method; CAT was determined by UV absorption; Chl was determined by spectrophotometry; and MDA was determined by the thiobarbituric acid method.

Net growth of plant height: The heights H_0_, H_7_ and H_14_ (cm) from the stem base to the terminal bud were measured on days 0, 7, and 14 of the NaCl and NaHCO_3_ treatments. The net growth of plant height Δ*_i_* on day *i* after treatment was calculated as
Δ*_i_*= H*_i_*− H_0_
where H_0_ is the plant height on day 0 after treatment and H*_i_* is the plant height on day *i* after treatment (*i* = 7 or 14).

### 3.4. Statistical Analysis

SPSS 17.0 was used for variance analysis. Duncan’s test was used to test the difference between treatments, and Microsoft Excel 2010 was used for plotting and membership function analysis.

The membership function can comprehensively evaluate the plant stress resistance on the basis of multi-indices measurement, and is widely used to analyze the plant stress resistance.

If the index is positively related to the resistance, the membership function calculation formula is (1). If the index is negatively related to the resistance, the membership function calculation formula is (2), and the average score formula of the membership function is (3). A high U*_A_* value indicates strong resistance, and vice versa.
(1)$$UX_{i}=\frac{\left(X_{i}-X_{min}\right)}{\left(X_{max}-X_{min}\right)}$$
(2)$$UX_{i}=1-\frac{\left(X_{i}-X_{min}\right)}{\left(X_{max}-X_{min}\right)}$$
(3)$$U_{A}=\frac{\sum_{i=1}^{n}U(X_{i})}{n}$$
where, U (*X_i_*) is the membership function value of a concentration treatment for the ith index, *X_i_* is the measured value of a concentration treatment for the *i*th index, *X_max_* and *X_min_* are the maximum and minimum values measured for the *i*th index; U*_A_* is the average value of the membership function of a concentration treatment.

## 4. Discussion

NaCl and NaHCO_3_ are the most common neutral and alkaline salts in the soil of arid and semi-arid areas in northwest China [30]. When NaCl and NaHCO_3_ concentration in the soil is too high, it will affect the normal growth of plants and lead to metabolic disorders or even to death in serious cases. In this study, when the concentration of NaCl and NaHCO_3_ in the soil was less than 100 mmol·L^−1^, it could significantly promote the height growth of *V. tricolor*, and the promotion effect of NaCl is higher than that of NaHCO_3_, which indicated that a certain concentration of NaCl in the soil could be beneficial to the growth of *V. tricolor*, but that when it became too high, the growth of *V. tricolor* was adversely affected.

Under the NaCl and NaHCO_3_ treatments, SS, SP and Pro in *V. tricolor* leaves were higher than those under CK, indicating that under NaCl and NaHCO_3_ stress, *V. tricolor* maintains the osmotic potential of cells by actively accumulating osmotic regulatory substances such as SS, SP and Pro, thereby improving the osmotic pressure of cells and maintaining a strong ability to absorb and retain water [31]. In addition, the contents of SS, SP and Pro under NaHCO_3_ treatments were higher than those under NaCl treatments, indicating that the tolerance of *V. tricolor* to NaHCO_3_ stress was higher than that to NaCl stress, which is consistent with the results of Al-Farsi [31].

In the present study, it was found that SOD, POD and CAT first increased and then decreased after *V. tricolor* was treated with NaCl and NaHCO_3_, indicating that with an increase in NaCl and NaHCO_3_ concentration, the degree of cell-membrane lipid peroxidation was intensified. *V. tricolor* has been shown to eliminate reactive oxygen species (ROS) such as hydrogen peroxide and superoxide anions by improving the activity of antioxidant enzymes [8,32,33], so as to slow down the damage to cells caused by ROS [4]. In the present study, when the NaCl and NaHCO_3_ concentration was lower than 150 mmol·L^−1^, SOD, POD and CAT were higher than those under CK. Moreover, when the NaCl and NaHCO_3_ concentration was higher than 150 mmol·L^−1^, SOD and POD were lower than in CK, indicating that *V. Tricolor* could eliminate ROS due to cellular peroxidation by increasing the activity of antioxidant enzymes when the NaCl concentration was lower than 150 mmol·L^−1^. The contents of osmoprotectants and antioxidant enzymes decreased with the increase in NaCl and NaHCO_3_ concentration. This may be because the metabolic regulation system and antioxidant enzyme system would fail when the NaCl concentration was higher than the range that *V. tricolor* would be able to adjust to; the defense system was unable to maintain a higher level, thereby inhibiting the growth and development of *V. tricolor* [9].

The chloroplast is the main organelle involved in photosynthesis in plants. Under NaCl and NaHCO_3_ stress, the structure and function of chloroplast are damaged, and chlorophyll is degraded due to the toxic effect of metal ions [34], which is positively correlated with the degree of ion stress in plants [35]. In this study, with the increase in NaCl and NaHCO_3_ concentration, the Chl increased first and then decreased. When the concentration was higher than 100 mmol·L^−1^, Chl under NaHCO_3_ treatments was higher than that under NaCl treatments, indicating that the degree of stress under NaHCO_3_ exposure was lower than that under NaCl exposure. The increase in Chl under low NaCl concentrations may be a physiological response of *V. tricolor* to actively adapt to stress. With the increase in NaCl and NaHCO_3_ concentrations, Chl decreased significantly, which was due to the increase in chloroplast enzyme activity and the promotion of Chl-*b* decomposition under high NaCl under NaHCO_3_ stress [36].

MDA is the product of cell membrane peroxidation, and its content is negatively correlated with plant stress resistance [37]. Therefore, MDA can be used to evaluate plant stress resistance. In this study, it was found that MDA increased with the increase in NaCl and NaHCO_3_ concentration. When the concentration was lower than 50 mmol·L^−1^, the degree of NaCl stress was lower than that of NaHCO_3_ stress; and when the concentration was higher than 50 mmol·L^−1^, the degree of NaCl stress was higher than that of NaHCO_3_ stress. This indicated that the damage to the plasma membrane system of *V. tricolor* caused by NaCl was greater than that caused by NaHCO_3_, which was related to electrolyte extravasation due to cell membrane damage. The cell membrane is the first barrier against external NaCl ions entering plant cells and plays an important role in the physiology of plant NaCl and NaHCO_3_ resistance [9]. When cell tissues of *V. tricolor* were subjected to ion stress, the large number of free radicals led to membrane lipid peroxidation [38]. The integrity and function of the cell membrane were damaged, and the stability of the cell membrane was reduced, resulting in a large amount of protoplasm overflow and macromolecular substances in cells spilling out through the damaged parts, which would affect the normal metabolism of *V. tricolor* and result in cell death in severe cases [37,38,39].

Plant growth can be considered the external performance of plant as a result of its physiological and biochemical processes as well as the phenotypic characteristics that can be directly observed. The growth of plants under adverse conditions is positively correlated with their resistance; if plants grow well under adverse conditions, they have high resistance [40].

In this study, it was found that the plant height growth under NaCl treatments was higher than under NaHCO_3_ treatments when the treatment concentration was lower than 50 mmol·L^−1^, and in both cases was higher than under CK, indicating that a low NaCl and NaHCO_3_ concentration could promote the growth of *V. tricolor*. When NaCl and NaHCO_3_ concentration was higher than 50 mmol·L^−1^, the height growth of *V. tricolor* decreased significantly, and the plant height growth under NaHCO_3_ treatments was higher than that under NaCl treatments, indicating that the degree of NaCl stress was greater than that of NaHCO_3_ stress [41].

The stress resistance of plants is the result of a comprehensive action of physiological and biochemical indices. The membership function analysis performed in this study showed that the maximum NaCl and NaHCO_3_ concentrations that *V. tricolor* can tolerate were 150 mmol·L^−1^ and 200 mmol·L^−1^, respectively. The growth of *V. tricolor* became restricted if the concentration exceeded these thresholds, which might be due to the simultaneous increase in Na^+^ and Cl^−^ concentrations with the increase in NaCl concentration, which has a dual ion stress effect. Studies showed that Cl^−^ participated in the photolysis and oxygen release reaction of water in the photosystem II of photosynthesis, and could improve the osmotic pressure of cells and the hydration of plant tissues [42]. However, plants needed less Cl^−^, while too much Cl^−^ would cause ion toxicity and inhibit plant growth; in addition, when the Na^+^ concentration in the soil was too high, Na^+^ would replace Ca^2+^ on the cell membrane, leading to loopholes in the cell membrane and exosmosis of protoplasts in the cell, which would change the ion species and concentrations. Furthermore, due to a large amount of Na^+^ flowing into the cell, the structure and function of protease and some other enzymes would be affected, which would destroy the metabolism of the cell, cause chloroplast destruction, and reduce plant photosynthesis [43]. NaHCO_3_ also had the dual stress effects Na^+^ and pH. The increase in pH in the soil would destroy the physiological function of plant roots, making it difficult for plants to absorb water and nutrients, and cause structural damage to plant roots. At the same time, high soil pH also affected the effectiveness of nutrient elements in the soil and impeded the uptake of metal ions such as Ca^2+^, Mg^2+^, Mn^2+^ and Fe^2+^, which in turn affected the growth of plants [16,44]. In this study, according to a comprehensive ranking of the results, when NaCl and NaHCO_3_ concentrations were less than 50 mmol·L^−1^, the degree of stress under the NaCl stress on *V. tricolor* was lower than that under the NaHCO_3_ stress; and when the concentration was higher than 50 mmol·L^−1^, the degree of stress under the NaCl treatment was higher than that under the NaHCO_3_ treatment. Although NaHCO_3_ imposed dual effects from Na^+^ and pH, the pH of the NaHCO_3_ solution in each treatment was 8.3; therefore, the pH stress caused by the NaHCO_3_ did not increase with the increase in NaHCO_3_ concentration. Therefore, at the same concentration of Na^+^, the stress from pH was weaker than that from Cl^−^, which differs from the results of Zhang et al. [40], who studied the stress of cucumber under NaCl and NaHCO_3_.

## 5. Conclusions

NaCl and NaHCO_3_ significantly affected the osmoprotectants, antioxidant enzymes, photosynthetic pigment, MDA and plant height growth of *V. tricolor*. On day 14 after treatment, SS, SP, Pro, CAT, POD, SOD, Chl and plant height growth showed a trend of first increasing and then decreasing with the increase in NaCl and NaHCO_3_ concentration, while the MDA content showed an increasing trend. Membership function analysis showed that the maximum concentrations of NaCl and NaHCO_3_ that *V. tricolor* was able to tolerate were 150 mmol·L^−1^ and 200 mmol·L^−1^, respectively. Beyond these concentrations, osmoprotectants and antioxidant enzymes were seriously affected, Chl degradation was intensified, the photosynthetic system was seriously damaged, and the growth of *V. tricolor* was negatively affected.

## Figures and Tables

**Figure 1 plants-12-00178-f001:**
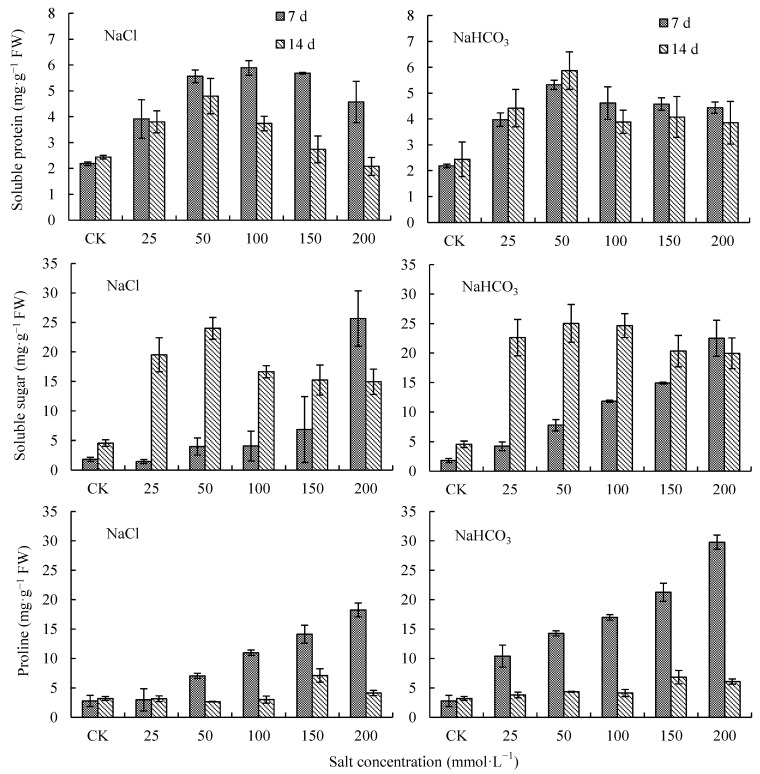
Protein, soluble sugar and proline in *V. tricolor* leaves under NaCl and NaHCO_3_ treatments.

**Figure 2 plants-12-00178-f002:**
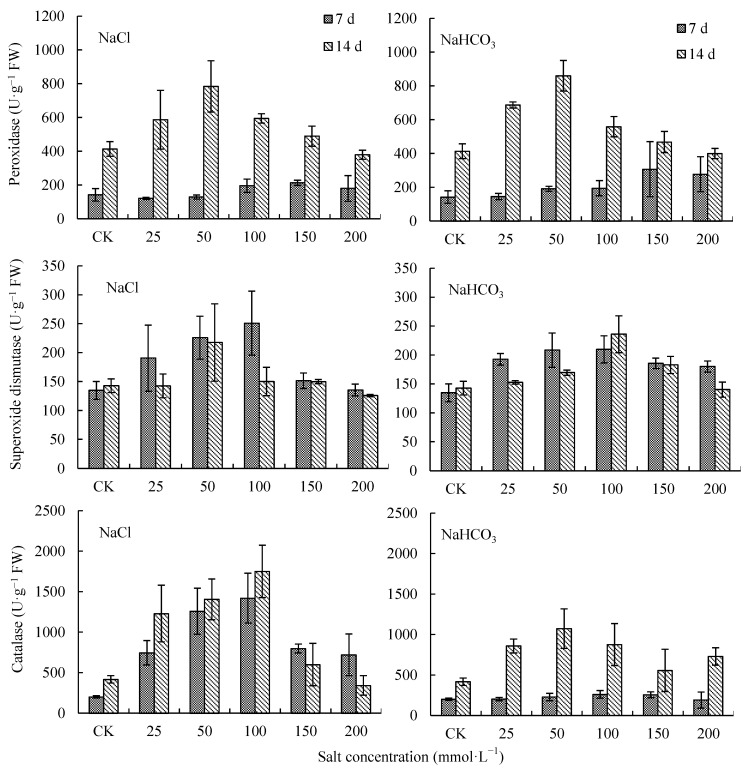
Superoxide dismutase and catalase in *V. tricolor* leaves under NaCl and NaHCO_3_ treatments.

**Figure 3 plants-12-00178-f003:**
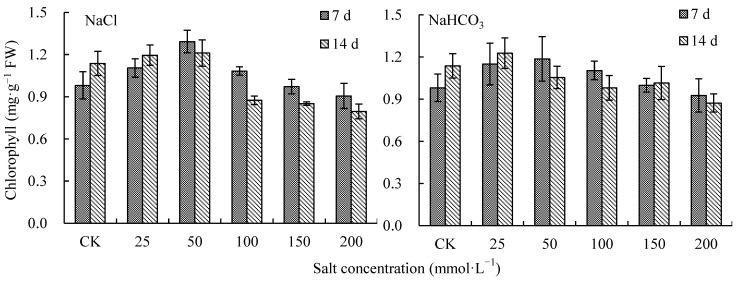
Photosynthetic pigments in *V. tricolor* leaves under NaCl and NaHCO_3_ treatments.

**Figure 4 plants-12-00178-f004:**
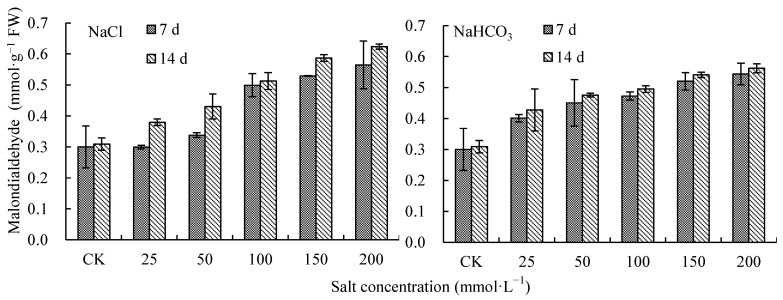
Malondialdehyde in *V. tricolor* leaves under NaCl and NaHCO_3_ treatments.

**Figure 5 plants-12-00178-f005:**
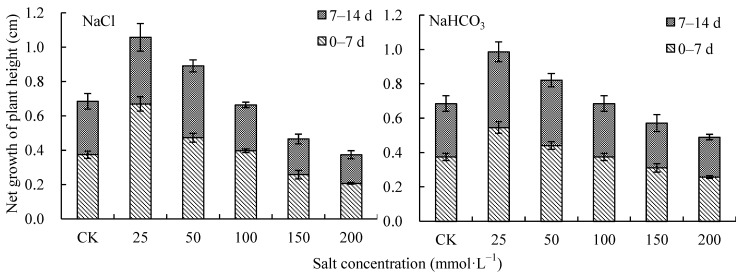
Height growth of *V. tricolor* under NaCl and NaHCO_3_ treatments.

**Table 1 plants-12-00178-t001:** Membership function scores of *V. tricolor* under NaCl and NaHCO_3_ treatments.

Treatment	Salt Concentration (mmol·L^−1^)	SS	SP	Pro	CAT	POD	SOD	Chl	MDA	Δ*_i_*	U*_A_*	Ranking
CK	0	0	0.110	0.144	0.069	0.088	0.170	0.808	1.183	0.471	0.338	9
NaCl	25	0.730	0.454	0.115	0.630	0.432	0.660	0.927	0.999	1.000	0.661	3
	50	0.949	0.716	0.000	0.755	0.841	0.834	0.963	0.877	0.757	0.744	1
	100	0.590	0.437	0.080	1.000	0.448	0.221	0.184	0.681	0.424	0.452	7
	150	0.521	0.173	1.000	0.182	0.229	0.219	0.129	0.504	0.134	0.343	8
	200	0.507	0.000	0.333	0.000	0.000	0.000	0.001	0.415	0.000	0.140	10
NaHCO_3_	25	0.882	0.617	0.259	0.367	0.641	0.245	1.000	0.884	0.895	0.643	4
	50	1.000	1.000	0.383	0.519	1.000	0.399	0.601	0.770	0.655	0.703	2
	100	0.981	0.477	0.331	0.378	0.373	1.000	0.429	0.723	0.455	0.572	5
	150	0.770	0.527	0.935	0.153	0.184	0.518	0.509	0.615	0.289	0.500	6
	200	0.717	0.433	0.736	0.240	0.008	0.097	0.145	0.528	0.135	0.338	9

**Table 2 plants-12-00178-t002:** Details of the NaCl and NaHCO_3_ treatment concentrations.

Salt Concentration (mmol·L^−1^)	Pretreatment					Treatment		
						Day 1	Day 7	Day 14
0 (CK)	0	0	0	0	0	0	0	0
25	0	25	25	25	25	25	25	25
50	0	25	50	50	50	50	50	50
100	0	25	50	100	100	100	100	100
150	0	25	50	100	150	150	150	150
200	0	25	50	100	150	200	200	200

## Data Availability

Not applicable.

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
