# Peer review of "Growth and Physiological Response of Viola tricolor L. to NaCl and NaHCO3 Stress"

_plants, 2023, doi:10.3390/plants12010178_

Round 1

Reviewer 1 Report (Previous Reviewer 1)

Dear Authors, thank you for the clarification and addition of the text excerpts. You managed to convince me, but not because there is a need to look for species that will serve to clean up degraded areas, but because of the promising results You obtained. I reviewed them again, and at the rather high doses you used, V. tricolor performed quite well. It seems to be promising species of tolerance close to the tolerance of halophytes. 

Author Response

Thank you, the attachment is response comments, please see it.

Reviewer 2 Report (Previous Reviewer 2)

Dear Author

Please follow the citation procedure. References without year or number need to be changed.

In the MS, it is written that, we hypothesize that the growth, physiological and biochemical characteristics of V. tricolor under NaCl and NaHCO3 stress are different, and the highest tolerance concentration of V. tricolor under NaCl and NaHCO3 stress is different with the increase of salt concentration. 

At first, it was not straight forward, but then we knew the mechanism of NaCl and NaHCO3 is different. Please verify this.

Please verify the crop husbandry protocol - 30 plants per replicate, 3 replications, sown in bowl. Not clear, please provide photos for self explanatory.

I found statistical analysis as novel

Author Response

Thank you, the attachment is response comments, please see it.

Reviewer 3 Report (Previous Reviewer 3)

Please check the spelling, eg:

lines 50 and 53: "," must be replaced by "."

line 228: "Peroxidase" must br replaced by "peroxidase"

Author Response

Thank you, the attachment is response comments, please see it.

This manuscript is a resubmission of an earlier submission. The following is a list of the peer review reports and author responses from that submission.

Round 1

Reviewer 1 Report

The experiments presented in this article are well described and presented, although I cannot find any justification for this research and the choice of this species. Viola tricolor is not a typical species for alkaline soils, saline soils, it is not an important economic plant. In my opinion, the low novelty disqualifies this article. I am really sorry, but the standard physiological and biochemical methods used for measurements on a non-model species do not bring much new to science.

Reviewer 2 Report

Dear Authors

Please explain the following for more clarity.

1. What is the significance of membership function analysis, how the threshold was obtained by that method has to be explained in statistics 

2. Similarly, what is the weightage given for each traits in comprehensive ranking - and what is the basis for that.

3. Please indicate your hypothesis and objective - both are missing

4. Please indicate what is the major conclusion from your study.

5. Why the author took two different salinity causing agent and compare it, both are having different mode of action.

6. How these two factor affect perennial plant, because it has its own resilience to salinity (based on literature survey).

7. Explain novelty - contradicting from 71-76 (line number).

8. What is the response of the crop to soil salinity reaction.

9. Please check the labels in the figures

10.

2.

Reviewer 3 Report

This manuscript could be an interesting contribution to the knowledge of physiological response of Viola tricolor to NaCl and NaHCO3 stress. However, I would like to point out some constraints:

General remarks:

1)     The Introduction and Discussion must be improved with other references about the studied   osmoprotectants in other plants namely Violaceae.

2)     The experimental design must be clarified: eg. line 96”each repetition” or “each treatment”; the “Pretreatment” referred in the table  in the table 1 is not referred in the text.

3)     English style and spelling needs to be improved

Other specific remarks:

1)     The species must have associated the author: Viola tricolor L.;

2)     According to POWO (https://powo.science.kew.org/) this species my be annual or subshrub.

3)     The subtitles “2.2.1” and “2.2.3” are not the are not the most suitable: eg. “experimental material”? ;